# Pilot Programme for Keratoconus Screening and Management in Adolescents with Down Syndrome

**DOI:** 10.3390/diagnostics15060683

**Published:** 2025-03-10

**Authors:** Barry Power, Kirk A. J. Stephenson, Diana Malata, Conor Murphy, Barry Quill, Fiona McGrane, Eleanor Molloy, William Power

**Affiliations:** 1Royal Victoria Eye and Ear Hospital, D02 XK51 Dublin, Ireland; 2Department of Ophthalmology, Royal College of Surgeons, D02 YN77 Dublin, Ireland; 3Department of Paediatrics, Tallaght University Hospital, D24 NR0A Dublin, Ireland

**Keywords:** keratoconus, Down Syndrome, screening, Pentacam, cornea

## Abstract

**Background/Objectives**: To screen a group of adolescents with Down Syndrome (Trisomy 21) for keratoconus and assess the feasibility of setting up a national screening service. **Methods**: Twenty-seven patients with Down Syndrome between 9 and 18 years of age attended our pilot keratoconus screening clinic. We recorded demographics, medical history, risk factors, best-corrected distance visual acuity, clinical examination results and corneal tomography results. The presence of keratoconus was confirmed by one of three corneal specialists based on clinical and tomographic findings. Tomographic analysis included zonal Kmax, thinnest point, inferior–superior asymmetry (IS Values), Belin/Ambrosio deviation value (BAD-D) and anterior and posterior elevation maps. **Results**: Early keratoconus was detected on tomography in 8 out of 54 eyes (15%) at the first review. These eyes were listed for crosslinking. The mean age of diagnosis was 14.6. Corneas in the Down Syndrome screening group were thinner and steeper (mean central corneal thickness (CCT) 479 µm vs. 536 µm and mean Kmax 49.2D vs. 45.8D, respectively) than healthy, age-matched controls from the literature. **Conclusions**: Fifteen percent of eyes (5 out of 27 patients) screened had tomographic evidence of keratoconus requiring treatment at their first review. We found an increased incidence of keratoconus in European individuals with Down Syndrome. Screening this vulnerable, high-risk population with corneal tomography can diagnose early keratoconus and enable corneal crosslinking to safely and effectively stabilise the disease. We advocate tomographic keratoconus screening for individuals with Down Syndrome in their mid-teens.

## 1. Introduction

Keratoconus is a progressive form of corneal ectasia in which gradual thinning leads to increasing irregular astigmatism and biomechanical weakness, leading to symptomatic visual loss [1,2]. Further corneal thinning may lead to a rupture in Descemet’s membrane (corneal hydrops), corneal scar formation and lasting reduction in visual function. The global prevalence of keratoconus is unclear and depends on the diagnostic criteria and population sampled. Individuals of Asian origin have the highest prevalence, and the highest incidence is between 20 and 30 years of age [3]. Risk factors for the development of keratoconus include atopy (e.g., asthma, eczema and allergic conjunctivitis), habitual eye rubbing and intellectual disability [4,5]. Two recent European estimates reported prevalence rates of 0.23 and 0.52 per 100,000, respectively [6,7].

Many visually significant ophthalmic issues are more common in Down Syndrome (Trisomy 21) than in the general population, including refractive errors (mostly hyperopia and astigmatism), strabismus (with esotropia being more common than exotropia), juvenile cataracts, nystagmus, retinal dystrophy, fundus depigmentation and choroidal sclerosis [8,9,10,11,12,13]. These conditions may cause amblyopia in the paediatric age group, the treatment of which (i.e., occlusion therapy) may be limited by behavioural factors. Keratoconus is also overrepresented in Down Syndrome populations, with prevalence varying widely in the literature (5.5 to 15%) due to differences in sample selection and diagnostic criteria [14,15,16,17,18,19,20]. The increased incidence of keratoconus in Down Syndrome is postulated to be related to several factors, including increased concomitant atopic conditions, associated collagen disorders (including Down’s arthropathy) and habitual eye rubbing [19,21,22].

Ireland has the second highest age-standardised incident rate of Down Syndrome globally, and it is estimated that there are about 7000 individuals with Down Syndrome in Ireland at present [23,24]. Corneal transplantation can be challenging in these individuals for a variety of different reasons (e.g., cooperation with post-operative drops, persistent eye rubbing and suture complications) [25]. Early-stage keratoconus can be successfully stabilised with corneal collagen crosslinking (CXL), which may prevent blinding complications, preserving vision and reducing the need for corneal transplantation [26,27]. Keratoconus typically presents in adolescence, progresses for 10–20 years and stabilises in the third or fourth decade of the individual’s life [28]. However, by age 35, significant corneal blindness may already be present in Down Syndrome patients [25]. There is evidence to show that reduced visual acuity contributes to the severity of intellectual disability and the development of dementia, both of which are common features in those with Down Syndrome [29,30]. The ideal age to screen for keratoconus and to offer treatment has not been established in Down Syndrome. This study reports on a pilot screening programme set up to assess children and adolescents with Down Syndrome for evidence of early/form fruste keratoconus that may be amenable to treatment or close monitoring. We aim to assess the feasibility of setting up a national screening programme for the disease and determine the optimum age and methodology for screening.

## 2. Materials and Methods

Patients in the Down Syndrome screening cohort were recruited from a clinic for children with Down Syndrome (Tallaght University Hospital, Dublin, Ireland). This clinic is a multi-disciplinary clinic that manages the often complex needs of paedatric and adolescent individuals with Down Syndrome in Ireland. This clinic did not previously have any formal links with an ophthalmology service. Patients attending the clinic were invited for an examination in a tertiary referral ophthalmic centre. Adolescents between the ages of 9 and 18 years were recruited, with their guardian’s consent. Exclusion criteria for participation in this study were intellectual disability other than Down Syndrome (e.g., autism or other genetic syndromes), forms of mosaic Down Syndrome, significant non-keratoconus corneal pathology (e.g., corneal dystrophy) or previous corneal surgery. Our research ethics board (REB) assessed the study and waived the need for formal REB review for this project considering its audit format. This paper reports on audit data collected after the first 3 years of the screening service. The SQUIRE checklist for quality improvement studies was used to ensure comprehensive reporting in writing this report [31].

Patients were clinically examined and the following clinical details were recorded: past medical history, risk factors (e.g., eye rubbing, eczema, asthma and other atopic conditions), best-corrected distance visual acuity (VA) and corneal tomography (Pentacam, Oculus Gmbh, Germany). Where possible, two 25-3D scans were taken from each eye to confirm the reliability of the data [32]. These scans were analysed by three expert corneal specialists (B.P., C.M. and W.J.P.) and the scan with the best quality data was used for clinical decision-making. Tomographic indices used to screen for the presence of keratoconus were as follows: zonal Kmax >48 dioptres (D), superior–inferior asymmetry index (IS Values) >1.4, Belin/Ambrosio deviation value (BAD-D) >1.6 and posterior elevation >10 at the thinnest point [33,34,35,36]. When all data were available, we defined early keratoconus as cases exceeding 3 or more abnormal tomographic parameters. A final diagnosis was then confirmed by a corneal specialist after a clinical exam. Management decisions were based on tomography and clinical exam and were made by one of three consultant corneal specialists following the algorithm outlined in our previous publication [37].

For those requiring intervention, treatment options considered included corneal collagen crosslinking (CXL) or penetrating keratoplasty (PKP). CXL was performed using the epithelium-off accelerated protocol by a clinical nurse specialist or an ophthalmologist [38]. Some patients were suitable for treatment under local anaesthesia, but most cases were performed under general anaesthesia. A bandage contact lens was used post-operatively for one week to minimise discomfort and consequent eye rubbing. Patients were discharged with a prescription for tapering topical steroids (Gutt prednisolone 1% QDS) and antibiotics (Gutt Ciprofloxacin QDS) and reviewed 1 week post-operatively. All patients (+/− guardians) consented to the examination and any treatment or intervention. PKP was considered cautiously due to the early stage of keratoconus and the high risk of transplant failure in DS patients.

Statistical tests were performed using SPSS statistics version 20 (IBM Corporation, Armonk, NY, USA). Though the study is underpowered by the nature of Down Syndrome as a rare genetic disorder, basic comparison with normative data from the published literature was possible using comparison of means tests (i.e., the one-sample *t*-test).

## 3. Results

Twenty-seven individuals (54 eyes) with Down Syndrome were included in the keratoconus screening group, and these patients were examined over a 3-year period (2020–2023). The mean age of the patients recruited was 13.8 years (range 9–18) and the male-to-female ratio was 13:14. The risk factor profiles of the screened patients are outlined in Table 1 and Table 2.

These data contrasted with normative paediatric data from Hashem et al., who reported tomography from 432 healthy children (432 eyes) [39]. This group had high baseline astigmatism, presumably the indication for corneal tomography. Most individuals in the adolescent Down Syndrome screening cohort were cooperative for the Pentacam examination (23 out of 27), but visual acuity data were limited by differing levels of intellectual disability and degree of refractive error. The demographics of the two groups were similar, with the normal control group having a mean age of 13.5 years (range 3–18). Table 3 compares the tomographic data between our screened Down Syndrome cohort and the healthy control group [39]. Our Down Syndrome screening cohort showed thinner and steeper corneas than the healthy age-matched controls (mean CCT 479 µm vs. 536 µm [*p* < 0.01], mean Kmax 49.2 vs. 45.8D [*p* < 0.01] respectively), as demonstrated in Table 3.

The mean BAD-D score was 2.5 in the Down Syndrome screening cohort and keratoconus was diagnosed in 8 out of 54 eyes (15%) and 5 out of 27 patients (19%). These eyes were listed for intervention (8 CXL). Figure 1 demonstrates the outcome of the first clinical review.

No patients had a PKP as diagnosed keratoconus was in the early stages and suitable for CXL. Treated patients were followed up for a mean of 11 months (range 3–34 months). There were no complications following CXL and no progression was noted in any treated eye (defined as an increase in astigmatism of >1D or a >1D increase in zonal Kmax over 1 year). Table 4 compares the tomographic data of the screening group who received treatment with those who did not.

## 4. Discussion

Individuals with Down Syndrome often present to ophthalmic services with advanced, corneal blindness due to keratoconus and individuals in this group are poor candidates for transplantation [25]. Acute hydrops is seen frequently in this vulnerable group [40] and individuals with Down Syndrome and keratoconus are overrepresented in studies of acute corneal hydrops [41]. Ideally, the diagnosis of keratoconus must be established at an earlier stage to prevent avoidable sight loss. A high proportion (82%) of our screened cohort required intervention or close specialist follow-up.

The link between keratoconus and Down Syndrome is well documented [17,18,20]. This keratoconus propensity is likely multifactorial (i.e., a high rate of atopy, habitual eye rubbing, environmental factors (pollutants, allergens and UV light exposure), intellectual disability and possible genetic factors (e.g., loci at 21q21.3 and 21q22.3) [42]. Early diagnosis and treatment are especially important in this cohort; however, the optimal age to screen patients has not been established. These data contrasted with normative paediatric data, with steeper (Kmax) and thinner (CCT) corneas being found in adolescents with Down Syndrome than in healthy, age-matched controls [39]. Notably, the prevalence of tomographic astigmatism was higher in the healthy control group [39] than in our Down Syndrome group. A high level of corneal astigmatism is the most common indication for tomographic assessment. In total, 46% of the patients in the Hashem cohort had astigmatism levels greater than 3D—we would assume that this was the indication for the Pentacam tomography investigation. The fact that K readings and CCTs were significantly steeper and thinner in our Down Syndrome screening group, despite the lower levels of astigmatism, highlights the unique corneal morphology found in people with Down Syndrome that leads to a higher risk of developing keratoconus. It also emphasises the advantage of performing advanced assessments such as corneal tomography, which provides substantial amounts of clinically relevant data with a non-invasive approach rather than retinoscopy, which has a more limited predictive value for keratoconus. Considering the greater prevalence of astigmatism in the control cohort, the Down Syndrome cohort would have even more significantly different tomographic parameters compared to a truly normal paediatric control cohort.

As mentioned previously, studies reporting on rates of keratoconus in Down Syndrome vary widely, in most part due to different diagnostic criteria and investigations performed. For example, Fimiani et al. collected data on a group of 157 Italian children with Down Syndrome and reported a keratoconus rate of 0% [9]. In this study, keratoconus was assessed using slit lamp biomicroscopy and retinoscopy alone, so the true prevalence of keratoconus in Down Syndrome is likely to be significantly underestimated. Studies utilising corneal pachymetry and topography may also underestimate keratoconus prevalence in comparison with more in-depth measures such as tomography. This allows for automatic application of keratoconus prediction algorithms (e.g., Belin/Ambrosio deviation value).

Compared with other studies on Down Syndrome populations utilising corneal tomography, our Down Syndrome screening group tomography data show similarly steep and thin corneas. Alio et al. found a mean K2 of 47.37D and an average CCT of 503 µm in a group of Down Syndrome individuals with an average age of 14 years (range 3 months to 60 years) [17]. Hashemi et al. screened individuals with Down Syndrome with an age range of 10–30 and found a mean Kmax of 49.04D and a keratoconus prevalence of 12.39% [20]. Of the 28 patients diagnosed with keratoconus, 24 were in the 10–20 years old category. Their one-year follow-up data showed similar rates of disease progression in comparison with keratoconus in a non-Down Syndrome adolescent/paediatric population. Our results mirror both these studies, despite the different patient populations with different ethnic backgrounds (White Western European vs. Arabic Middle Eastern ethnicity).

We contrasted the findings of our paediatric screening cohort with nine Down Syndrome adults with keratoconus (average age 35, range 22–48) who were referred to our clinic. Eight out of eighteen eyes (44%) in this adult group were listed for intervention after their first appointment (1 PKP, 7 CXL) in comparison with 21% in the paediatric cohort. Despite concerns (e.g., eye rubbing and drop installation) there were no post-treatment complications in the adult cohort. Four out of eighteen eyes in the adult group exhibited keratoconus too advanced for safe/effective CXL and patient-related factors prohibited safe corneal transplantation. It is clear that earlier intervention with CXL is of paramount importance in this population. Using these permanent functional and structural findings, we can support the utility of early (i.e., adolescent) tomographic screening for keratoconus and early stabilising treatment (i.e., CXL) in at-risk populations, especially children with Down Syndrome.

Contact lenses are a mainstay of visual correction in keratoconus but may not be a suitable option for many individuals with Down Syndrome due to compliance issues and the risk of keratitis. The risk of progression of thinning to levels unsuitable for CXL must also be considered as corneal transplantation has a poorer prognosis in the Down Syndrome population [25]. Epithelium-on CXL may be a consideration in cases of thin corneas considering the risks of PKP in this population [43]. CXL may even be beneficial in the presence of a small (≤3 mm) apical corneal scar to prevent further hydrops episodes and the formation of a larger central scar [40]. Conservative measures, like the avoidance of eye rubbing, are less likely to be successful in groups with intellectual disability and complex needs. Previous evidence supports early screening and intervention (i.e., CXL) under anaesthesia or sedation in patients with severe intellectual disability that prohibits awake assessment [37].

Corneal tomography is the gold standard for the diagnosis of keratoconus. Assessment of corneal thickness and posterior elevation are crucial indicators of early-stage ectasia. The ability to undergo accurate corneal tomography may vary with age and the degree of intellectual disability. Despite imaging a relatively young Down Syndrome cohort (average age 13 years), most individuals (85%) were cooperative for tomography. Retinoscopy is a sensitive screening tool, but its value for staging keratoconus and monitoring its progression is limited [44,45]. In the early stages of keratoconus (i.e., forme fruste), retinoscopy is limited in its power to detect subtle abnormalities. Tomography is therefore our preferred screening method for a high-risk group like Down Syndrome as it is sensitive to early, clinically undetectable disease. The relatively modest levels of astigmatism (mean 2.12D) detected in our Down Syndrome group also highlights the need to perform tomography rather than relying on screening based on high levels of detected astigmatism. If screening were dependent on retinoscopy only, vulnerable patients would suffer permanent corneal blindness due to inadequate follow-up. Early (i.e., adolescent) tomographic assessment by a corneal specialist can identify early, serious but treatable corneal ectasia to prevent long-term morbidity for this vulnerable patient group.

Ireland has an estimated annual Down Syndrome incidence of 1 in 546 births with an annual birth rate of 11.3/1000 population [46]. Thus, we estimate that there are approximately 100 children born each year with Down Syndrome in Ireland. Previously published evidence shows that people with Down syndrome are at significant risk of developing keratoconus compared to the normal population, and our findings support the use of tomography-based corneal screening for keratoconus in this population. Opportunistic engagement with centralised Down Syndrome clinics enables efficient screening to prevent corneal blindness in this vulnerable group, who also face numerous life-long health concerns (e.g., congenital heart disease, dementia, diabetes and intellectual disability). We believe that an increase in access to tomography nationally makes tomography-based screening of this cohort feasible. We believe that active screening for keratoconus in Down Syndrome fulfils the nine Wilson criteria for a screening programme [47].
Condition: It should be suitable for screening.Test: There should be a suitable test for the condition.Treatment: There should be effective and accessible treatment for the condition.Evidence: There should be high-quality evidence that the screening programme is effective.Benefit: The potential benefit of the screening programme should outweigh the potential harm.Health care system: It should be able to support the screening programme.Social and ethical issues: They should be considered.Cost–benefit issues: They should be considered.

There are some limitations to this study. Tomographic data can vary in accuracy, particularly in groups where patient cooperation (e.g., fixation and head position/stability) can be limited. Visual outcomes were difficult to assess given the lack of data and varying use of refractive correction (i.e., glasses/contact lenses/no correction). Incorporation of intellect-appropriate visual acuity assessments (e.g., preferential looking, number, shape or matching optotypes) may help to more accurately document VA. Due to the relative rarity of Down Syndrome and the fact that this study was carried out during the restrictions enforced during the COVID-19 pandemic, the study population is relatively small despite Ireland having one of the largest Down Syndrome populations per capita in the world. This may have induced a degree of selection bias, and larger, more inclusive studies may provide more reliable data. The control group may not be a true representation of the healthy paediatric population as they had high baseline astigmatism, presumably the indication for corneal tomography [39]. Despite these limitations, we feel that our data are robust and highlight the needs of this vulnerable patient population.

## 5. Conclusions

Individuals with Down Syndrome are a vulnerable group and have a higher incidence of many medical and ophthalmic issues [48]. Sight loss is a devastating experience for any individual but is even more debilitating in those with complex needs. It exacerbates intellectual disability and predisposes individuals to earlier dementia [29,30,48,49]. Early keratoconus can be detected and managed, with relative ease, through tomographic screening and CXL. We have shown that a significant number (8 out of 54 eyes) of screened eyes required crosslinking after the first screening exam. Furthermore, the procedure was carried out safely in all individuals with no procedure-related complications. Corneal crosslinking is a very cost-effective intervention [50,51]. Our data, unsurprisingly, highlight the fact that symptomatic adults with Down Syndrome referred to the clinic with suspected or previously confirmed and untreated keratoconus have more advanced disease and a higher rate of intervention than a younger screened population. Additionally, safe intervention may not be possible in older age groups; thus, proactive screening of at-risk Down Syndrome adolescents/teenagers should be sought. We strongly advocate tomographic screening for this population. We believe the optimal age for this to be the early teens to maximise the ability to undergo accurate tomography and minimise disease progression that can occur in early adulthood.

## Figures and Tables

**Figure 1 diagnostics-15-00683-f001:**
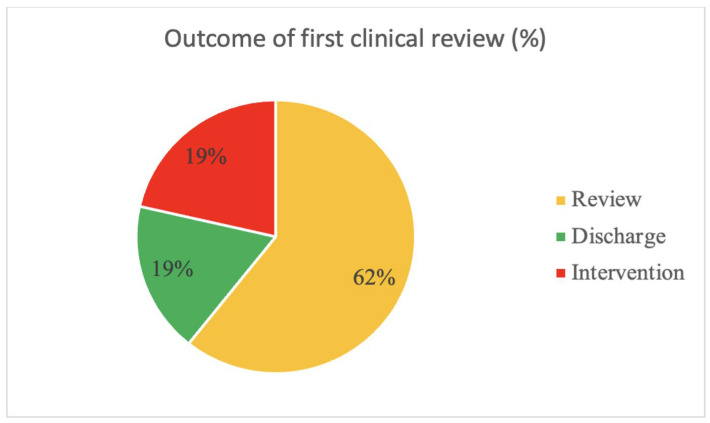
Outcome of the first clinical review.

**Table 1 diagnostics-15-00683-t001:** Risk Factor Profile of Screened Patients

Risk Factor	
Asthma	11%
Eczema	11%
Eye Rubbing	30%
Hay Fever/Atopy	26%

**Table 2 diagnostics-15-00683-t002:** Number of Risk Factors Present in Screened Patients.

No. of Risk Factors	
0	40%
1	48%
2	11%
3	1%
4	0%

**Table 3 diagnostics-15-00683-t003:** Comparison of Tomography of Screening Group vs. Controls

	Screening		Control [39]		*p*-Value
	Mean	95% CI	Mean	95% CI	
K1 (D)	45.2	44.7–45.7	42.3	42.2–42.5	0.01
K2 (D)	47.3	46.7–48.0	45.1	44.9–45.3	0.01
Kmax (D)	49.2	47.7–50.6	45.8	45.6–46	<0.01
Astigmatism (D)	2.1	1.5–2.7	2.8	2.7–2.9	0.3
CCT (µm)	479	455–502	536	532–542	<0.01

**Table 4 diagnostics-15-00683-t004:** Comparison of Tomography in Treated and non-Treated Individuals

	**Treatment**		**No Treatment**		***p*-Value**
	**Mean**	**95% CI**	**Mean**	**95% CI**	
K1 (D)	44.7	43.8–45.7	45.3	44.8–45.8	0.44
K2 (D)	48.2	46.1–50.3	47.1	46.5–47.4	0.45
Kmax (D)	52.5	46.8–58.1	48.2	47.5–48.8	0.04
Astigmatism (D)	3.5	1.2–5.8	1.7	1.4–1.9	0.04
BAD-D	6.7	1.0–12.2	1.9	1.5–2.2	0.35
CCT (µm)	489	459–518	477	466–511	0.92

## Data Availability

Clinical data are unavailable due to data protection regulations, but anonymised data are available from the corresponding author upon reasonable request to barryjapower@gmail.com.

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
