# Peer review of "Pilot Programme for Keratoconus Screening and Management in Adolescents with Down Syndrome"

_diagnostics, 2025, doi:10.3390/diagnostics15060683_

Round 1
Reviewer 1 Report
Comments and Suggestions for Authors
Dear authors,
Thanks for the manuscript. Here my comments
Abstract:
1. Line 12: rephrase in 54 eyes of twenty sevent-patients (not indivduals)
2. Line 18-19: add the age of patients which were diagnosed with Keratoconus
Introduction:
Consider to shorten the introduction into 3 paragraph. For example the first paragraph of current version (line 32-44) can be deleted or be shortened and merged with second/third paragraph.
Methods:
1. Line 105-106: add the type of Pentacam scan protocol used (consider to add reference: DOI: 10.3390/jcm14020439). Please also, details if quality scans different from "OK" have been considered.
2. Line 115-117: the comparison with the other study (Hashem et al.) should be removed from methods and moved to discussion to comment your findings with those reported in literature
3. Line 119-127: in case of Keratoconus detection, patients were straight listed for CXL or was before assessed any progression?
4. Line 128-133: The review of cohort adult patients could be confusing. Maybe would be better or to include it, making it clear also in the title of manuscript, or to removed it, as accordingly to the title, methods, results, the main focus is on adolescent, not in adult too.
Results:
1. It would be beneficial to add a table where are reported the demographic, age, sex, and numerical parameters differences between the patient who were positive for Keratoconus accordingly to the screen and those who were negative.
2. Line 150-174: it is mentioned "screening cohort" but is it not clear now if it is related to adolescent or the adult cohort mentioned. Add adolescent to make it clearer.
3. During the follow-up of adolescent screening cohort no patients experienced any progression of Pentacam parameters which could be suggestive for keratoconus progression?
Author Response
The authors wish to thank the reviewers for their time and expertise in reviewing this manuscript. We have included point-by-point responses to your comments below. We are confident that the manuscript is now more relevant and portrays our intended message in a clearer and more concise way.
Reviewer 1
Dear authors,
Thanks for the manuscript. Here my comments
Abstract:
1. Line 12: rephrase in 54 eyes of twenty sevent-patients (not indivduals)
Author response: Thank you. This has been amended to ‘patients’ rather than ‘individuals.’
Line 18-19: add the age of patients which were diagnosed with Keratoconus
Author response: Added line 20 (mean age is 14.6).
Introduction:
Consider to shorten the introduction into 3 paragraph. For example the first paragraph of current version (line 32-44) can be deleted or be shortened and merged with second/third paragraph.
Author response: We have significantly shortened the introduction as requested. Thank you for pointing this out.
Methods:
1. Line 105-106: add the type of Pentacam scan protocol used (consider to add reference: DOI: 10.3390/jcm14020439). Please also, details if quality scans different from "OK" have been considered.
Author response: Where possible, and in the majority of cases, scan quality was “OK”. 4 scans were included that had lower quality. We decided to include these to keep our data applicable to real world clinical practive where not all scans are perfect. The protocol used was the 25 3D (line 101)
Line 115-117: the comparison with the other study (Hashem et al.) should be removed from methods and moved to discussion to comment your findings with those reported in literature
Author response: This reference has been moved to the discussion as suggested. The presence of baseline astigmatism in the control group has been added to the limitations.
- Line 119-127: in case of Keratoconus detection, patients were straight listed for CXL or was before assessed any progression?
Author response: This DS population was considered high risk for progression based on previous experience. This was also a cohort of young patients (adolescents/young adults) and KCN detected at a very young age is at very high risk of progression. Waiting for progression increases the risk of corneal scarring developing and increases corneal astigmatism – particularly in groups with behavioural challenges. Our experience with CXL in DS children and adults has been positive, with no serious adverse effects noted (PMID: 35244626). We prefer early intervention with CXL in this cohort when KCN is detected.
- Line 128-133: The review of cohort adult patients could be confusing. Maybe would be better or to include it, making it clear also in the title of manuscript, or to removed it, as accordingly to the title, methods, results, the main focus is on adolescent, not in adult too.
Author Response: To avoid confusion, we have removed this from the methods section and only compared this with the adolescent study cohort in the discussion section.
Results:
1. It would be beneficial to add a table where are reported the demographic, age, sex, and numerical parameters differences between the patient who were positive for Keratoconus accordingly to the screen and those who were negative.
Author response: That is a good suggestion, I have added a Table 4.
Line 150-174: it is mentioned "screening cohort" but is it not clear now if it is related to adolescent or the adult cohort mentioned. Add adolescent to make it clearer.
Author response: Thank you. We have added ‘adolescent’ as suggested. Having removed mention of the adult cohort from the methods, this should prevent confusion.
During the follow-up of adolescent screening cohort no patients experienced any progression of Pentacam parameters which could be suggestive for keratoconus progression?
Author response: The patients treated with CXL have been followed up in the clinic and have not shown evidence of progression to date (average follow up 11 months) (defined as defined as an increase in astigmatism of >1D or a >1D increase in zonal Kmax over 1 year).
Reviewer 2 Report
Comments and Suggestions for Authors
The authors address a critical clinical issue: keratoconus screening in patients with Down syndrome, providing valuable insights into early detection and management of these patients. However, several significant concerns need to be addressed to enhance the study’s impact and applicability.
1. In methods, the diagnostic criteria for early keratoconus in this cohort should be more explicitly defined. Comparing these criteria with established diagnostic standards would strengthen the credibility and generalizability of the findings.
2. The sample size is relatively small, therefore, a discussion on potential selection bias and its possible effects on the findings would be beneficial.
3. The authors mentioned in the methods section that for patients diagnosed with keratoconus, the treatment options include corneal collagen cross-linking or penetrating keratoplasty. However, in the results section, it appears that all patients defined as having keratoconus only underwent corneal collagen cross-linking. Were there no patients who required penetrating keratoplasty?
4. Including a participant flow diagram would enhance the transparency of the study.
5. In cases of early-stage keratoconus, aside from corneal collagen cross-linking, the use of rigid gas permeable (RGP) lenses may also help to manage the progression of corneal pathology to some degree and for a certain duration. Is this approach applicable to patients with Down syndrome? This aspect warrants further discussion in the Discussion section.
6. Corneal topography systems may not be standard equipment in certain regions or healthcare facilities. It is recommended to consider and evaluate complementary screening approaches that are better suited for resource-constrained environments, where access to advanced diagnostic imaging technologies is often limited or unavailable.
Author Response
The authors wish to thank the reviewers for their time and expertise in reviewing this manuscript. We have included point-by-point responses to your comments below. We are confident that the manuscript is now more relevant and portrays our intended message in a clearer and more concise way.
The authors address a critical clinical issue: keratoconus screening in patients with Down syndrome, providing valuable insights into early detection and management of these patients. However, several significant concerns need to be addressed to enhance the study’s impact and applicability.
- In methods, the diagnostic criteria for early keratoconus in this cohort should be more explicitly defined. Comparing these criteria with established diagnostic standards would strengthen the credibility and generalizability of the findings.
Author response: Lines 98 to 111 have been rewritten to better clarify our diagnostic criteria. I have added several citations which support the tomographic parameters outlined. As this was a real world clinic, a final decision on diagnosis and treatment rested with a corneal specialist and followed a clinical exam taking into account the tomorgraphy.
- The sample size is relatively small, therefore, a discussion on potential selection bias and its possible effects on the findings would be beneficial.
Author response: Thank you for highlighting this. We have added the following on lines 300: ‘the study population is relatively small despite Ireland having one of the largest Down Syndrome populations per capita in the world. This may have induced a degree of selection bias and larger, more inclusive studies, may provide more reliable data’
- The authors mentioned in the methods section that for patients diagnosed with keratoconus, the treatment options include corneal collagen cross-linking or penetrating keratoplasty. However, in the results section, it appears that all patients defined as having keratoconus only underwent corneal collagen cross-linking. Were there no patients who required penetrating keratoplasty?
Author response: All patients in this young screening cohort who were diagnosed with keratoconus had good vision and early stage disease. However, they were considered to be at high risk of progression to corneal blindness. Thus, these patients (19%) were treated with CXL rather than PKP.
The following has been added to lines 166: ‘No patients had a PKP as diagnosed keratoconus was early stage and suitable for CXL.’
- Including a participant flow diagram would enhance the transparency of the study.
Author response: The following has been added to lines 109: ‘Management decisions were made by one of 3 consultant cornea specialists following the algorithm outlined in our previous publication [36].’
- In cases of early-stage keratoconus, aside from corneal collagen cross-linking, the use of rigid gas permeable (RGP) lenses may also help to manage the progression of corneal pathology to some degree and for a certain duration. Is this approach applicable to patients with Down syndrome? This aspect warrants further discussion in the Discussion section.
Author response: The discussion on line 237 has been expanded as follows: ‘Contact lenses are a mainstay of visual correction in keratoconus but may not be a suitable option for many individuals with Down Syndrome due to compliance issues and risk of keratitis.’
- Corneal topography systems may not be standard equipment in certain regions or healthcare facilities. It is recommended to consider and evaluate complementary screening approaches that are better suited for resource-constrained environments, where access to advanced diagnostic imaging technologies is often limited or unavailable.
Author Response: This is discussed on line 249. While more readily accessible screening tools have some utility, that is not the purpose of the current study. We advocate that children/adolescents with T21 should have early specialist assessment in centres where tomography is available. This can detect significant disease early in life and enable CXL before corneal blindness occurs.
We have clarified our position on lines 262: ‘Early (i.e., adolescent) tomographic assessment by a corneal specialist can identify early serious but treatable corneal ectasia to prevent long-term morbidity for this vulnerable patient group.’
Reviewer 3 Report
Comments and Suggestions for Authors
Dear Authors,
I carefully read this relevant research, which undoubtedly contributes to the early diagnosis of keratoconus in a vulnerable population. While the data presented are valuable, as they reflect the higher prevalence that we overlook due to the lack of screening programs, there is confusion based on how the article is written, starting from the title. For example, the title states that it is a pilot program for keratoconus screening in adolescents with Down syndrome. However, as one progresses through the article, it becomes clear that all age groups were included, and treatments were applied. This means it is not just a screening program but also a program evaluating treatment outcomes. For this reason, I recommend rewriting the article by including only the screening part or changing the title and sections to reflect everything the study involves. By definition, screening is the testing of a person or group of people for the presence of a disease or other condition, so please make your article reflect that.
Best regards.
Author Response
The authors wish to thank the reviewers for their time and expertise in reviewing this manuscript. We have included point-by-point responses to your comments below. We are confident that the manuscript is now more relevant and portrays our intended message in a clearer and more concise way.
Dear Authors,
I carefully read this relevant research, which undoubtedly contributes to the early diagnosis of keratoconus in a vulnerable population. While the data presented are valuable, as they reflect the higher prevalence that we overlook due to the lack of screening programs, there is confusion based on how the article is written, starting from the title. For example, the title states that it is a pilot program for keratoconus screening in adolescents with Down syndrome. However, as one progresses through the article, it becomes clear that all age groups were included, and treatments were applied. This means it is not just a screening program but also a program evaluating treatment outcomes. For this reason, I recommend rewriting the article by including only the screening part or changing the title and sections to reflect everything the study involves. By definition, screening is the testing of a person or group of people for the presence of a disease or other condition, so please make your article reflect that.
Author response: Thank you for assessing our manuscript.
We have altered the title to: ‘Pilot Programme for Keratoconus Screening and Management in Adolescents with Down Syndrome.’
We have also removed the comment regarding adult patients from the methods and only referred to this in the discussion. Hopefully this improves the clarity of the manuscript and is true to the amended title.
Round 2
Reviewer 1 Report
Comments and Suggestions for Authors
Dear authors,
Thanks for the updated manuscript and the reply to comments.
Just minor consideration: line 116-118. Correctly have added the type of scan of Pentacam, no reference has added. It would be beneficial to consider add a reference, as there is a reason on why in case of Keratoconus 25-3D should be preferred over other scan protocols provided by Pentacam software (ref: https://doi.org/10.3390/jcm14020439)
Author Response
Just minor consideration: line 116-118. Correctly have added the type of scan of Pentacam, no reference has added. It would be beneficial to consider add a reference, as there is a reason on why in case of Keratoconus 25-3D should be preferred over other scan protocols provided by Pentacam software (ref: https://doi.org/10.3390/jcm14020439)
Reference added
Reviewer 3 Report
Comments and Suggestions for Authors
NA
Author Response
Many thanks for your time.